# Effects of Jet Milling Pretreatment and Esterification with Octenyl Succinic Anhydride on Physicochemical Properties of Corn Starch

**DOI:** 10.3390/foods11182893

**Published:** 2022-09-17

**Authors:** Lidong Wang, Xiaoqiang Li, Fei Gao, Shilin Liu, Yanchun Wu, Ying Liu, Dongjie Zhang

**Affiliations:** 1College of Food Science, Heilongjiang Bayi Agricultural University, Daqing 163319, China; 2Daqing Center of Inspection and Testing for Agricultural Products and Processed Products Ministry of Agriculture and Rural Affairs, Heilongjiang Bayi Agricultural University, Daqing 163319, China; 3Department of National Coarse Cereals Engineering Research Center, Heilongjiang Bayi Agricultural University, Daqing 163319, China

**Keywords:** corn starch, jet milling, OSA esterification, structure, physicochemical properties

## Abstract

(1) Background: In this study, aiming at the problems of low efficiency and high energy consumption in the esterification reaction of OSA and starch, the jet milling technology was used to pretreat corn starch and starch raw materials with different pulverization strengths were obtained by controlling the speed of the classifier. (2) Methods: The starch obtained under different classification speeds was modified by esterification with OSA. Using CLSM, FTIR, XRD, NMR, FTIR, XPS, and other technologies, the modification effect was verified, and the physical and chemical properties of J-OSA-Starch such as DSC, RVA, transparency, and emulsifying properties were determined. (3) Results: Jet milling pretreatment significantly reduced the particle size of corn starch and improved the reaction efficiency and degree of substitution during esterification with OSA. After pretreatment, the corn starch granules were broken, and the relative crystallinity was significantly reduced. CLSM, FTIR, XPS, and NMR confirmed the esterification of corn starch with OSA, which increased with increasing crushing strength. The thermodynamic properties and viscosity of J-OSA-starch decreased with the increase in the classification speed. Jet milling pretreatment enhanced the clarity, emulsifying activity, and emulsifying stability of OSA-modified corn starch. (4) Conclusions: Jet milling pretreatment can effectively increase the esterification efficiency of starch and OSA. Therefore, jet milling can be used as a pretreatment to improve the esterification of starch OSA and produce modified starch for industrial applications.

## 1. Introduction

Starch is a micro or submicron particle formed by amylose, amylopectin, fat, proteins, minerals, salts, and nanocrystals (orthorhombic and/or hexagonal). Starch can be more widely used in the food industry by physical and chemical modification [1]. Studies showed that OSA esterification treatment changes starch properties [2]. The native starch is changed from hydrophilic to hydrophobic by replacing the octenyl molecule during the esterification with OSA, making the whole molecule amphiphilic [3].

Although starch with OSA can be used as an additive in food, its application is limited due to the low substitution degree and low reaction efficiency [4]. At present, three common strategies are applied to esterify starch with OSA. The first one, dry starch is dissolved and dispersed in organic solvents for the reaction, but this method is expensive and not environmentally friendly. The second is to mix starch with lye, adjust the moisture, spray into the acid anhydride, mix well, and then heat for the reaction, which has the advantages of low cost and low pollution, but poor uniformity, high synthesis temperature, and serious product pyrolysis. The third one, starch, is pretreated by extrusion, microwave, radiation, ball milling, or other physical methods or biological enzymes to destroy the crystalline structure of starch granules and improve the reaction activity of starch granules or increase the specific surface area to promote the esterification reaction. However, due to some technical problems related to starch and equipment, the industrial application of these physical methods is limited [5].

Jet milling technology is used to generate high-speed airflow through the nozzle of the compressed gas after drying and purification and drive the particles to move at high speed in the crushing cavity so that the particles are crushed by impact collision, friction, shearing, etc. The qualified particles are collected by the collector, while the coarse particles that do not meet the requirements are returned to the crushing chamber to continue crushing until the required fineness is reached and collected. Compared with the ball mill and vibrating mill, the jet mill has the characteristics of low energy consumption, short pulverization time (1 h), and low pulverization temperature (25 °C). In the crushing cavity, high-speed airflow created by the jet milling technology is driven by the high-speed movement of the particles. As a result, the particles are subjected to impact collision, friction, shear, and other crushing effects, and the crushed particles are classified according to classifier rotational speeds [6]. As a method of mechanical force crushing, jet milling gradually reduces the size of the particles and increases the specific surface area. The jet milling changes the internal structure, physicochemical properties, and chemical reaction sites of starch [7]. Due to the low substitution and reaction efficiency during the OSA-modification process of starch, pretreatments may help in improving the OSA-modification efficiency. In order to explore the reaction efficiency and substitution degree of micronized corn starch and OSA obtained at different classification speeds, changes in structural and physicochemical properties of jet milling-pretreated and OSA-modified corn starch were evaluated using different methods and techniques.

The previous research of our group showed that jet milling could break starch granules and significantly increase the specific surface area of starch. This change means that more groups that react with OSA are exposed, which can effectively increase the esterification efficiency. Then the United Nations Food and Agriculture Organization and the World Health Organization (FAO/WHO) stipulated that the OSA ester could be used in food, but the maximum processing capacity of octenyl succinic anhydride should not exceed a 3% level, and the influencing factors such as temperature and pH are sufficient in the esterification modification of different starches. In this study, starch obtained at different classification speeds was subjected to an esterification reaction with OSA, and the effect of pulverization degree on esterification efficiency was studied.

## 2. Materials and Methods

### 2.1. Materials

Normal corn starch was obtained from Heilongjiang Longfeng Cron-developing Co., Ltd. (Suihua, China). The starch composition was determined according to the methods of the Association of Official Analytical Chemists (AOAC). The normal corn starch (wet weight) contained 13% (*w*/*w*) moisture, 0.30% protein, 0.05% fat, and less than 0.1% ash. All other chemicals and reagents were of analytical grade and purchased from Chemical Company, Tianjin.

### 2.2. Jet Milling Pretreatment

A fluidized bed jet mill (Model LHL Fluid Bed Jet Milling Equipment, Powder Equipment Company, Weifang, Shandong, China) was used for pretreatment of corn starch, and the equipment parameters and pulverization conditions were as follows: 3 nozzles, draft fan flow rate of −15 m^3^/min, feed size less than 0.85 µm, pulverization time of 1 h, feed volume of 1 kg, jet mill temperature of 25 °C [8], and classifier rotational speeds of 1200, 1800, 2400, 3000, and 3600 rpm were used. The pretreated starch samples were marked as J1-starch, J2-starch, J3-starch, J4-starch, and J5-starch depending on the classifier rotational speed, respectively, and stored in sealed containers at 4 °C for further use.

### 2.3. Preparation of Octenyl Succinate Starch

A starch suspension with a concentration of 30% (*w*/*w*) was prepared by mixing 200 g of jet milling-pretreated starch sample (dry base) with distilled water. The pH of the starch milk was adjusted to 8.5 with a 2% NaOH solution. A total of 6 g of OSA was weighed, diluted three times with anhydrous ethanol, and added dropwise to the starch suspension within 1.5 h. The reaction pH was monitored and maintained at pH 8.5. The reaction was continued for 1.5 h under stirring. After the reaction, the pH was adjusted to 6.5 with a 2% HCl solution. Starch was washed 2 times using 90% ethanol and distilled water, followed by centrifugation. The samples were dried at 40 °C for 24 h, then ground and passed through a 100-mesh sieve [9]. Jet milling-pretreated and OSA-modified starch samples were marked as OSA-starch, J1-OSA-starch, J2-OSA-starch, J3-OSA-starch, J4-OSA-starch, and J5-OSA-starch.

### 2.4. Octenyl Succinate Starch Particle Size, Degree of Substitution, and Reaction Efficiency

#### 2.4.1. Particle Size

The particle size of starch was determined using a laser particle sizer (Bettersize 2000, Dandong Better Instrument Co. Ltd., Shandong, China) as described by Wang [10]. Starch samples were mixed with distilled water, stirred well, sonicated for 5 min, and then analyzed.

#### 2.4.2. Octenyl Succinate Starch DS and RE

OSA-starch was accurately weighed to 5.0 g (dry weight), 25 mL of 2.5 mol/L hydrochloric acid-isopropanol solutions were added, and the mixture was stirred continuously at room temperature for 30 min. A 50 mL of 90% (*v*/*v*) aqueous isopropanol solution was then added to the mixture and stirred for 10 min. The sample was transferred to a Buchner funnel and filtered, and the residue was washed with 90% isopropanol solution until no chloride ions could be detected with 0.1 mol/L AgNO_3_ solutions. The sample residue was dissolved in 300 mL of deionized water and heated in boiling water for 20 min, 2 drops of phenolphthalein indicator were added, and the temperature was maintained and rapidly titrated to pink with 0.1 mol/L standard NaOH solution [11].

DS calculation formula.
(1)$$DS=\frac{0.162\times (A\times M)/W}{1-\left[0.209\times (A\times M)/W\right]}$$
where *A* is the volume of the titrated NaOH standard solution, mL. *W* is the mass of the dry basis of the sample of OSA-starch, g. *M* is the concentration of the standard solution of NaOH, mol/L.

RE calculation formula was as follows:(2)$$RE=\frac{210\times DS\times m_{s}}{162\times m_{0}}\times 100\%$$
where 162 is the molar mass of the glucose residue, g/mol. A total of 210 is the molar mass of OSA, g/mol. *m_s_* and *m_o_* are the masses of starch and OSA-starch used in the reaction, g.

### 2.5. Microscopy Observation

#### 2.5.1. SEM

Appropriate amounts of starch samples were taken and spread evenly on a sample stage with conductive adhesive, followed by gold spray treatment, and placed under a scanning electron microscope (S-3400N, Hitachi Limited, Tokyo, Japan) for observation at magnifications of 2000× [12].

#### 2.5.2. CLSM

A sample of 0.5 g starch was mixed with 30 mL of deionized water, and the pH was adjusted to about 8.0 with 1 mol/L NaOH solution. After adding 1% (*w*/*w*) methylene blue (MB^+^) solution, the sample was stirred magnetically at room temperature for 24 h. The excess MB^+^ was washed out with methanol. Finally, the starch granules were mixed with a glycerol-water mixture (1:1, *v*/*v*). A drop of the starch suspension was placed on a slide, covered with a coverslip, and the overall distribution of octenyl succinate groups in the starch granules after fluorescent staining was observed (LSM800, Carl Zeiss AG, Oberkochen, Germany). The lens used in this study was 40×/1.25 oil, and the gas laser argon laser emitted at 514 nm [13].

### 2.6. XRD and FTIR

#### 2.6.1. XRD

Starch samples were analyzed following the step-scan method (BrukerD8, Advance Optics Solutions GmbH, Dresden, Germany), and the detection conditions were as follows: CuKα (1.5406) as the characteristic ray, 40 kV tube voltage, 100 mA current, 2θ interval of measurement angle from 4° to 60°, 0.02° step size, and 4°/min scanning speed [12]. Based on the intensity of diffraction peaks at different diffraction angles, the MDI jade software was used to calculate the starch crystallinity.

#### 2.6.2. FTIR

A sample of 3.0 mg starch was mixed with 300 mg of KBr powder, ground, and pressed into tablets. Samples were scanned at 400–4000 cm^−1^ and a resolution frequency of 4 cm^−1^ [14] (Nicolet6700, Thermo Fisher Scientific, Waltham, MA, USA).

### 2.7. XPS

An X-ray energy spectrometer (ESCALAB250Xi, Thermo Fisher Scientific, Waltham, MA, USA) was used for the elemental analysis of the starch surface. The test conditions were as follows: Al-Kα line of 1486.6 ev, a line width of 0.8 ev, a vacuum of 5 × 10^−9^ Pa, emission voltage of 15 KV, and power of 150 W. The C1s peak (binding energy of 284.6 eV) was used as a standard for energy correction, and the sample was dried to constant weight and filled with inert gas for sealing protection before testing [15].

### 2.8. Nuclear Magnetic Resonance H and C Spectra

#### 2.8.1. Nuclear Magnetic Resonance H Spectra

Maltose-free starch samples of 5 mg each were dissolved in DMSO-d_6_. A total of 2 mg of deuterated trifluoroacetic acid was added before analysis. 1H NMR spectral conditions (Bruker AVANCE III HD NMR 400M, Bruker Magnetic Resonance, Dresden, Germany) were as follows: analysis temperature of 30 °C, pulse angle of 30°, relaxation time of 10 s, and detection time of 2 s [16].

#### 2.8.2. Nuclear Magnetic Resonance C Spectra

Starch samples were dissolved in DMSO-d_6_ (containing tetramethylsilane TMS) and measured at 100.62 MHz, 25 ± 1 °C (Bruker AVANCE III HD NMR 400M, Bruker Magnetic Resonance, Dresden, Germany), using the chemical shift of TMS as a reference [17].

### 2.9. Thermodynamic Properties

Starch samples of 6.0 mg each (dry basis) were placed into an aluminum tray, and deionized water was added at 1:3 (*m*/*v*), sealed, and equilibrated at room temperature for 24 h. The temperature range was 20~100 °C with a constant heating rate of 10 °C/min (DSC800, PerkinElmer, Waltham, MA, USA) [18].

### 2.10. Pasting Characteristics

A starch sample of 3.5 g weight was dispersed in 30 mL of distilled water in an aluminum pot and mixed using a homogenizer, and analyzed by a rapid viscosity analyzer (RVA-Super4, Newport Scientific, Sydney, Australia) to obtain starch pasting characteristic values [19].

### 2.11. Transparency Measurement

The starch suspension was prepared at a concentration of 1% (*w*/*v*), heated in a water bath at 30 °C, 70 °C, and 100 °C in water, and then cooled to room temperature. The light transmission of the starch paste was measured using a UV spectrophotometer at a wavelength of 650 nm, and distilled water was used as a reference [19].

### 2.12. Emulsifying Ability and Emulsification Stability Index

A certain amount of starch sample was mixed with deionized water to prepare a 1.5% (*w*/*w*) starch emulsion, which was placed in a boiling water bath at 100 °C and stirred continuously for 20 min. The gelatinized starch emulsion was cooled to room temperature, 5% of peanut oil was added based on starch weight, homogenized for 1.5 min at 10,000 rpm using a high-speed homogenizer, and repeated twice. Aspirate 50 μL of the starch emulsion was mixed with 5 mL of 0.1% sodium dodecyl sulfate solution. The initial absorbance (*A*_0_) of the sample was measured by a UV spectrophotometer at 500 nm using a 0.1% concentration of sodium dodecyl sulfate solution as a blank, and the value obtained was the EA of the sample [20].

The starch sample was allowed to stand for 10 min, and its absorbance at 500 nm (*A*_10_) was determined again by a UV spectrophotometer. The emulsification stability index (ESI) was calculated using the following equation [21].
(3)$$ESI(\min)=\frac{A_{0}}{A_{0}-A_{10}}\times 20$$

### 2.13. Statistical Analysis

All experiments were carried out at least in triplicate, and the data were statistically analyzed and expressed as mean ± SD. Analysis of variance (ANOVA) and Duncan’s multiple range test with a confidence interval of 95% were performed using SAS software version 9.2 (SAS Institute, Inc., Cary, NC, USA).

## 3. Results

### 3.1. Octenyl Succinate Starch Particle Size, Degree of Substitution, and Reaction Efficiency

Changes in the particle size and specific surface area of corn starch after pretreatment and after OSA modification by jet milling at different classifier rotational speeds are shown in Table 1. The coarse end size (D_90_) was 20.06 μm, the median size (D_50_) was 13.43 μm, and the fine end size (D_10_) was 5.87 μm. At a classifier speed of 3600 rpm (J5-starch), the particle size parameters were reduced to 13.34 μm, 7.70 μm, and 2.08 μm, respectively. However, the specific surface area (Sw) increased from 0.237 m^2^/g for the native corn starch to 0.403 m^2^/g for the jet milling-treated corn starch at a classifier speed of 3600 rpm. It can be seen from Table 1 that the effect of OSA modification on starch particle size and specific surface area is not significant.

The degree of substitution and substitution efficiency of OSA-starch obtained after the esterification of corn starch with OSA without jet milling pretreatment were 0.0170 and 73.5%, respectively (Table 1). However, the degree of substitution and substitution efficiency of OSA-modified corn starch after pretreatment by jet milling were higher than those of the native corn starch and gradually increased with the increase in jet milling pretreatment classifier speed, to up to 0.0186 and 80.4%. This indicates that the pretreatment of corn starch with jet milling technology can effectively improve the efficiency of OSA modification of starch. This can be attributed to the fact that the crystalline structure and particle morphology of starch are destroyed during the pretreatment process, the crystallinity is reduced, the specific surface area is increased, and the reaction sites are increased, enhancing the substitution degree and reaction efficiency.

### 3.2. SEM and CLSM

Changes in the morphology and surface structure of starch granules were analyzed using a scanning electron microscope, and the microscope images are shown in Figure 1. As can be seen from Figure 1A1–F1, the size of the native corn starch granules is large with a smooth surface and randomly embedded with small holes. After OSA modification of the native corn starch, holes or depressions appeared on the surface of starch granules. The degree of granule fragmentation gradually increased, producing some tiny granules, and the particle size gradually decreased for corn starch samples pretreated with jet milling. This can be attributed to the fact that the mechanical force of jet milling produced starch particles with irregular morphology depending on the classifier speed. On the other hand, and as can be seen from Figure 1A2–F2, after OSA modification, depressions appeared on the surface of granules, and the surface of the particles became rough with an irregular shape [22].

CLSM is commonly used to study changes in the internal structure of starch granules. In this study, a methylene blue (MB^+^) fluorescent stain was used to specifically label the -COO^−^ of the OS group inside and on the surface of OSA-starch granules, thus observing the distribution of the OS group in OSA-starch. Figure 1A3–F3 shows CLSM images of the jet milling-pretreated and OSA-modified corn starch samples. The native corn starch did not show fluorescence (images are not shown). The fluorescence of OSA-modified starch without jet milling pretreatment appeared mainly at the periphery of the starch granules, indicating that OSA reacted on the surface of starch, and the OS groups were distributed on the surface of the starch granules. This can be attributed to the fact that the OSA agent is not easily soluble in water, and it is difficult for the OSA droplets to penetrate inside the starch granules, thus forming a stronger fluorescence on the surface of starch granules. Laser irradiation of OSA-starch pretreated by airflow crushing at different intensities revealed that the content of OS groups (-COO^−^) in OSA-starch was positively correlated with the degree of substitution, i.e., higher substitution showed stronger fluorescence intensity [23]. When the classifier rotational speed of jet milling treatment was 3600 rpm, it can be seen that the starch was broken into fine particles, the OSA binding sites to the starch increased, and the fluorescence increased and intensified. This indicates that jet milling pretreatment improved the esterification of the corn starch with OSA by enhancing the degree of OS substitution.

### 3.3. XRD and FTIR

#### 3.3.1. XRD

Figure 2 shows XRD spectra for the native and treated corn starch. The native corn starch showed a typical “A-type starch” with strong characteristic diffraction peaks around 15.8°, 17.1°, 18.2°, and 23.5°. It can be seen in Figure 2A. After the jet milling process, with the increase in the classification speed, although no new diffraction peaks were generated, the intensity of the diffraction peaks weakened with the increase in the rotation speed, which indicated that the jet milling did not change the crystalline morphology of ordinary corn starch, and still showed “A-type starch”, but pulverization reduces the crystallization degree of common corn starch. It can be seen in Figure 2B. In contrast, no new diffraction peaks were generated after the esterification with OSA. The degree of crystallinity for the J-OSA-starch samples decreased as the jet milling classifier speed increased, which can be attributed to the fact that the mechanical force of jet milling destroyed the internal crystalline structure of corn starch, the jet milling induced destruction of amylopection and average amylopection chain length, decreasing starch crystallinity, resulting in an increase in the non-crystalline region and a decrease in starch crystallinity, while the OSA-modified groups were mainly grafted in the non-crystalline region of the starch [24]. These results indicate that jet milling pretreatment significantly reduced the crystallinity of corn starch and enhanced the esterification of corn starch with OSA.

#### 3.3.2. FTIR

When the corn starch is refined by air jet milling, compared with the infrared spectrum of the original starch, as shown in Figure 3A, the peak positions of the main peaks have basically not changed, and no new peaks have appeared, indicating that no new peaks have been generated groups, but the intensity and width of some characteristic peaks changed. The intensity of the characteristic peak at 3422 cm^−1^ increases [25], indicating that the associative hydrogen bonds of starch molecules are broken by jet milling, the number of hydroxyl groups increases, the characteristic peak is narrowed, and the intensity weakens or disappears, indicating that the jet milling process is broken by strong mechanical force. The crystal structure of starch granules was transformed from an ordered structure to an amorphous structure.

Functional groups of organic compounds can perform selective absorption of infrared light. Therefore, FTIR can be used to qualitatively analyze the grafting of chemical groups on starch after starch esterification by OSA and to evaluate the effect of modification treatments on the ordered structure of starch crystalline, as well as molecular chain conformation and helical structure [26]. Figure 3 shows FTIR spectra for the native and treated corn starch. The native corn starch structure showed characteristic bands at 1640 cm^−1^ and 1420 cm^−1^. These bands are attributed to the bending vibration of H2O and the stretching vibration of the hydroxyl group, and the nearby bands at 3500–3000 cm^−1^ reflect the hydrated structure of the starch. Compared to the native corn starch, two new characteristic bands appeared at 1724 cm^−1^ and 1573 cm^−1^ for OSA-esterified starch. This is due to the substitution of the hydroxyl group on the starch molecule by the carboxyl group in OSA after the corn starch was modified by OSA esterification, resulting in the C=O stretching vibration and the generation of asymmetric vibrations of the carboxyl group in the starch, indicating successful synthesis of OSA-modified starch, which is consistent with results of Li [22]. The intensity of these two characteristic bands increased as the classifier speed of jet milling pretreatment increased. This indicates that both the pretreatment intensity and the number of moieties of OSA-starch are positively correlated with the intensity of the absorption bands, which is consistent with the results of Zhang [27]. These results indicate that jet milling pretreatment improved the OSA-corn starch esterification.

### 3.4. XPS

X-ray photoelectron spectroscopy (XPS) is an analytical technique for obtaining the distribution of different chemical states of elements on the surface of a sample by measuring the photon energy produced by the sample and is widely used to study changes in the structural and surface groups of modified starch [28]. Figure 4 shows the XPS spectra of the native, jet milling pretreated, and OSA-esterified corn starch. Distinct C1s and O1s chemical binding energy characteristic peaks can be seen at positions 284 eV and 533 eV, which can be attributed to the fact that starch is a polymeric polysaccharide composed of many glucose units of three elements, C, O, and H, linked by glycosidic bonds. In contrast to the native and jet milling pretreated corn starch, OSA-esterified starch showed a new Na peak at 1070 eV on the XPS spectrum, indicating that OS groups are introduced on the starch molecule, and the OSA-starch esterification was successful, which is consistent with the findings of Liu [29].

Table 2 shows the surface elemental composition of the native, jet milling pretreated, and OSA-esterified corn starch. The relative content of carbon and oxygen elements within the corn starch molecules changed slightly after the esterification treatment, which was caused by the substitution of the hydroxyl group on the starch molecule by the carboxyl group from OSA during the esterification process. Moreover, the content of sodium elements increased with the increase in jet milling pretreatment classifier speed. The XPS spectra indirectly also reflect that the jet milling pretreatment promoted the esterification of corn starch with OSA.

### 3.5. ^1^H-NMR and ^13^C-NMR

The ^1^H-NMR and ^13^C-NMR spectra for the native, jet milling pretreated, and OSA-esterified corn starch samples are shown in Figure 5, respectively. The ^1^H-NMR spectra revealed that the OSA groups were introduced into the molecular structure of corn starch during the esterification process. As the classifier speed of the jet milling pretreatment increased, the relative intensity of the spectral peak weakened, where the chemical shift of OH_2_ was located, and no obvious changes were found in peaks of other chemical shifts, indicating that the esterification reaction of jet milling-pretreated corn starch with OSA might mainly occur on the OH2. For the ^13^C-NMR spectrum, the chemical shifts of C12 and C13 on the unsaturated double bond were in the range of 110–150; the chemical shifts of C17 and C18 on the carboxyl and ester carbonyl groups were in the range of 155–185, but the overall esterification of the jet milling-pretreated corn starch with OSA was not obvious. These results illustrate the effect of jet milling pretreatment on the OSA modification at the level of starch molecular structure.

### 3.6. Thermodynamic Properties

From Figure 6, it can be seen that there is a large thermal absorption peak at 70 °C for the native corn starch, which decreased for the jet milling-pretreated and OSA-esterified corn starch. From Table 3, values of the Thermodynamic characteristics of the native corn starch are high, and after jet milling pretreatment, the thermodynamic characteristics values of OSA-esterified starch decreased as the frequency of the jet milling increased. This indicates that the molecules of native corn starch are closely arranged and crystalline, and after jet milling pretreatment, the starch granules and the internal molecular chains were broken due to mechanical force. Jet milling pretreatment reduced the relative crystallinity of starch, making it easier for starch to absorb heat, showing lower T_0_, T_P_, T_C,_ and enthalpy values. For the samples esterified by OSA, the thermal absorption peak at 70 °C was restored, which is consistent with the results of increased crystallinity and increased order by XRD and FTIR.

### 3.7. Pasting Characteristics

Changes in the pasting characteristics of the native and treated corn starch are presented in Table 4. The paste viscosity and paste temperature of corn starch modified by OSA esterification are higher than those of the native starch. Due to the introduction of octenyl succinic anhydride, the starch molecules with the same negative charge would repel each other [30]. This can be attributed to the introduction of macromolecular OSA groups into the starch after starch modification by OSA, which increased the relative molecular weight of the starch and thus led to an increase in the apparent viscosity of the starch paste. In contrast, the hydrogen bonds of OSA-modified starch were broken, and the macromolecular chains were damaged after pretreatment by jet milling, and the OSA groups were introduced into the starch, which increased the starch spatial site resistance and reduced the intermolecular interactions between starch molecules and increased the hydration and swelling of starch granules [31,32,33]. On the other hand, the peak viscosity, trough viscosity, final viscosity, pasting temperature, attenuation and regrowth value of the J-OSA-starch sample decreased with the increase in airflow crushing intensity. Compared with OSA-modified starch without jet milling pretreatment, the peak viscosity, trough viscosity, final viscosity, pasting temperature, and regrowth value of OAS-esterified and jet milling-pretreated starch decreased. Therefore, the final viscosity of OSA-modified starch with jet milling pretreatment decreased gradually with the increase in pretreatment classifier speed.

### 3.8. Transparency

From Figure 7, The transparency of OSA-modified starch paste increased significantly after jet milling pretreatment depending on classifier speed. Transparency of the OSA-modified starch did not change significantly at 30 °C, indicating that the starch was not gelatinized at low temperature, but only formed a turbid liquid of starch granules; with the increase in pasting temperature, the increase in paste transparency was still small at 70 °C. However, when the pasting temperature increased to 100 °C, the transparency of starch paste increased significantly and increased with the increase in jet milling pretreatment classifier speed. These phenomena may be attributed to the destruction of the crystalline structure and transformation to an amorphous form. Because of that, the water can enter the interior of starch granules, leading to increased transparency, which reduces the molecular weight of starch and increases the number of small molecules, reducing light scattering and refraction. After the OSA-modification of starch, the starch granules and molecular structure are changed; therefore, water molecules can penetrate the starch granules, and the starch granules swell and break easily, increasing the light transmission.

### 3.9. Emulsifying Ability and Emulsification Stability Index

From Table 4, emulsification and emulsion stability of jet milling-pretreated and OSA-esterified starch significantly improved compared to the native and OSA-esterified corn starch without jet milling pretreatment and more enhanced with the increase in milling pretreatment pulverization intensity. This can be attributed to the fact that OSA modification is a hydrophobic modification, which increases the effective contact sites between the starch and oil phases at the interface, forming stable emulsions [34]. The continuous film formed at the oil-water interface of oil-in-water emulsions becomes stronger and less likely to break with the increase in OSA substitution, weakening the tension at the oil-water interface and making it more difficult for the dispersed phase particles to coalesce and separate, improving the emulsion stability. With the increase in hydrophobic groups, the ability of esterified starch to reduce surface tension was increased; thus, the emulsification and emulsion stability of esterified starch were enhanced. The emulsification and emulsion stability of the J5-OSA-starch sample obtained by pretreatment with jet milling at a classifier frequency of 3600 rpm was the highest, which is consistent with the OS substitution results

## 4. Conclusions

The corn starch was treated by jet milling technology, and different pretreated starch samples were obtained according to the classification speed. It was found that the pretreatment of jet milling reduced the particle size and relative crystallinity of the starch, increased the specific surface area of the particles, and improved the reaction efficiency with OSA. Degree of substitution. FTIR, XPS, and NMR analyses confirmed that the esterification efficiency of cornstarch with OSA increased with increasing jet milling intensity. The physicochemical properties analysis showed that the thermodynamic and RVA parameters of the J-OSA-starch prepared by jet milling were significantly reduced, and the transparency and emulsification were significantly improved. Based on the obtained results, jet milling can be considered a promising pretreatment method to improve the efficiency of the esterification reaction of starch with OSA to produce OSA-modified starch for industrial applications. Further studies are required to investigate the application properties of jet mill pretreatment and OSA-modified starch.

## Figures and Tables

**Figure 1 foods-11-02893-f001:**
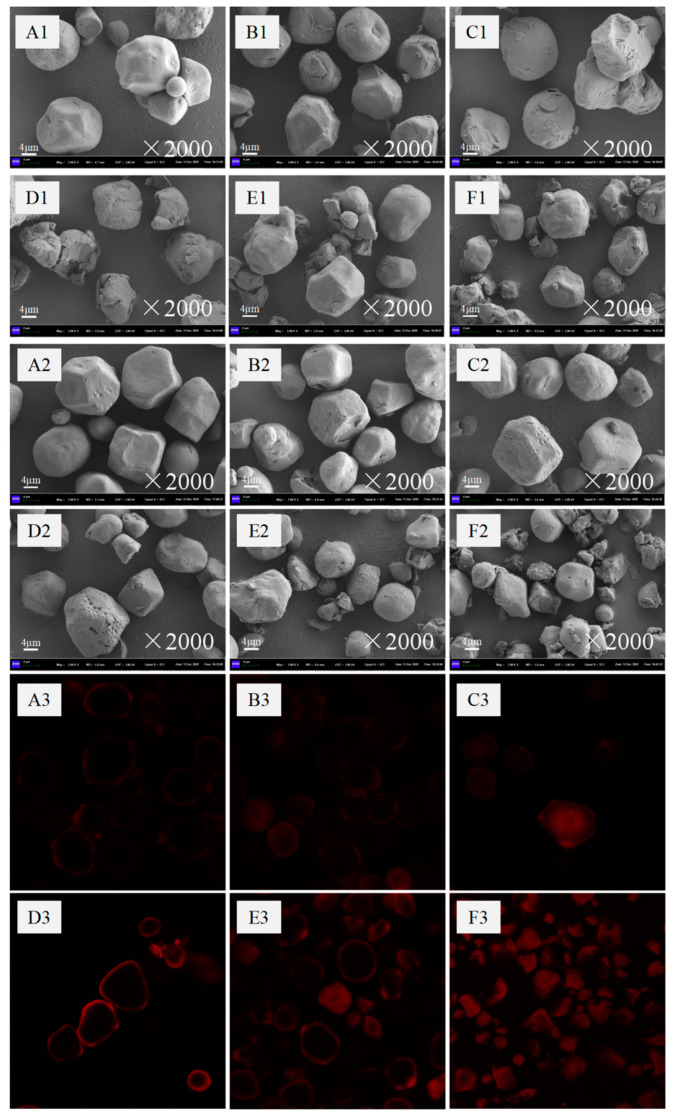
SEM and CLSM of the native and treated corn starch samples: **A1**–**F1** are the SEM images of native starch and **J1**–**J5** starch, respectively; **A2**–**F2** are the SEM images of OSA-starch and **J1**–**J5** OSA-starch, respectively; **A3**–**F3** are OSA-starch and **J1**–**J5** OSA-starch, respectively CLSM images.

**Figure 2 foods-11-02893-f002:**
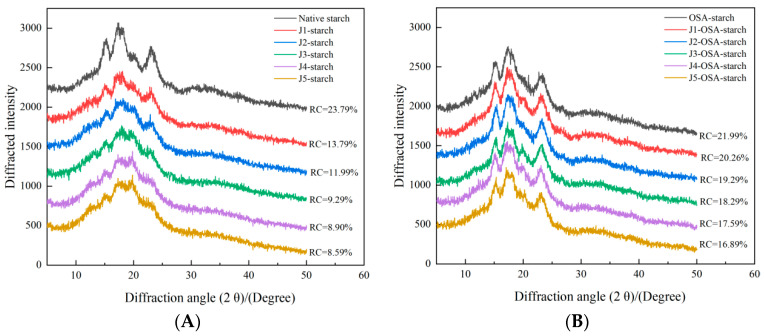
XRD patterns of J-starch and J-OSA-starch: (**A**), XRD patterns of J-starch; (**B**), XRD patterns of J-OSA-starch.

**Figure 3 foods-11-02893-f003:**
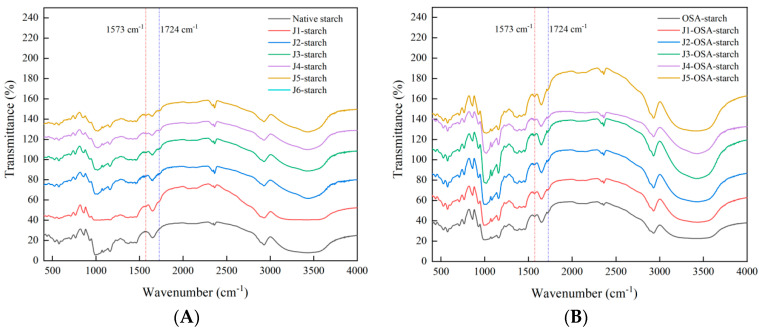
FTIR spectra of J-starch and J-OSA-starch: (**A**), FTIR patterns of J-starch; (**B**), FTIR patterns of J-OSA-starch.

**Figure 4 foods-11-02893-f004:**
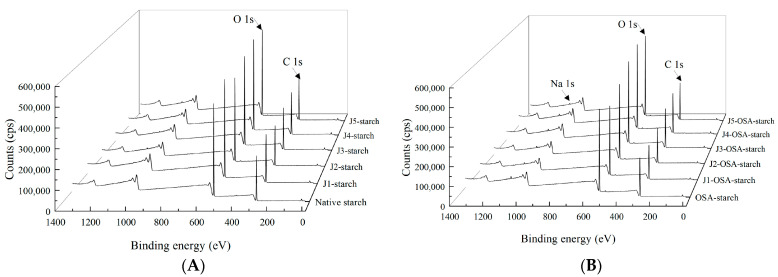
XPS analysis of J-starch and J-OSA-starch: (**A**), XPS patterns of J-starch; (**B**), XPS patterns of J-OSA-starch.

**Figure 5 foods-11-02893-f005:**
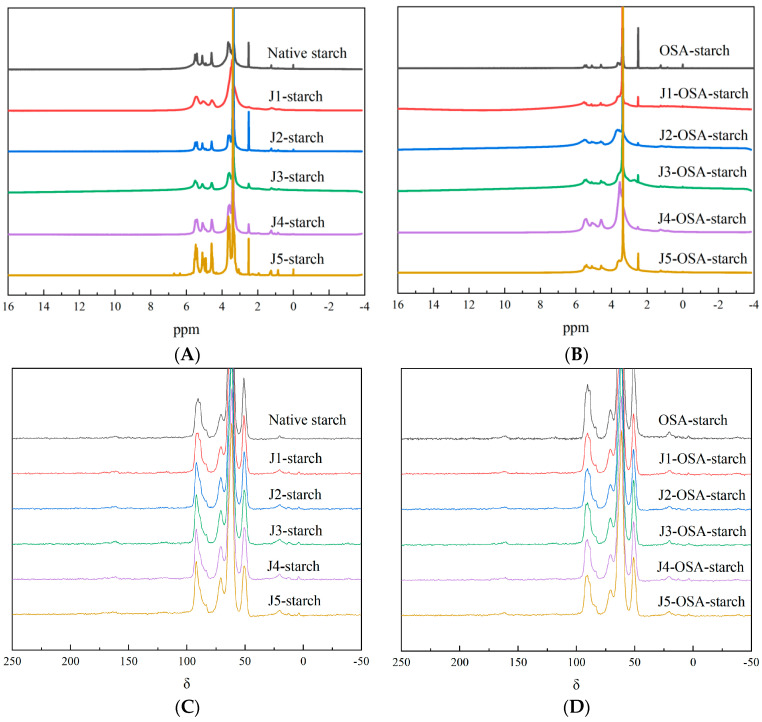
^1^H-NMR and ^13^C-NMR spectra of J-starch and J-OSA-starch: (**A**), ^1^H-NMR spectra of J-starch; (**B**), ^1^H-NMR spectra of J-OSA-starch; (**C**), ^13^C-NMR spectra of J-starch; (**D**), ^13^C-NMR spectra of J-OSA-starch.

**Figure 6 foods-11-02893-f006:**
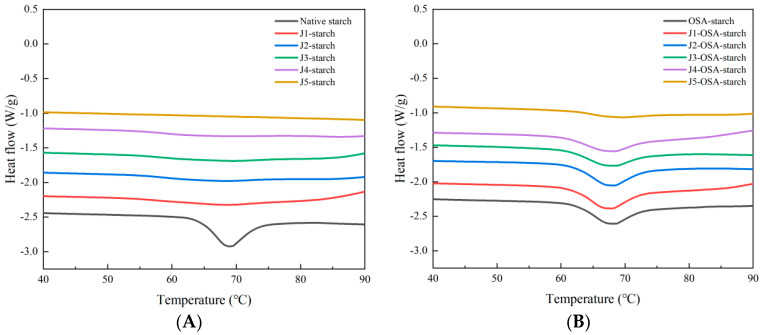
DSC spectra of J-starch and J-OSA-starch: (**A**), DSC spectra of J-starch; (**B**), DSC spectra of J-OSA-starch.

**Figure 7 foods-11-02893-f007:**
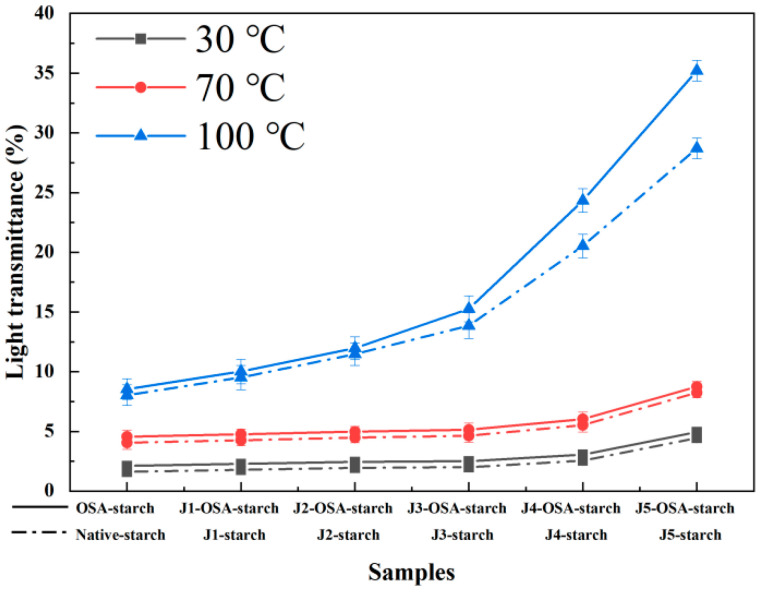
Transparency of the treated corn starch samples.

**Table 1 foods-11-02893-t001:** Particle size parameters, degree of substitution, and reaction efficiency of J-starch and J-OSA-starch.

Samples	D_10_ (μm)	D_50_ (μm)	D_90_ (μm)	Sw (m^2^/g)	DS	RE (%)
Native starch	5.87 ± 0.07 ^a^	13.43 ± 0.03 ^a^	20.06 ± 0.02 ^a^	0.237 ± 0.001 ^a^	——	——
J1-starch	4.78 ± 0.02 ^b^	12.53 ± 0.04 ^b^	19.21 ± 0.06 ^b^	0.264 ± 0.002 ^b^	——	——
J2-starch	4.08 ± 0.08 ^c^	11.59 ± 0.02 ^c^	18.29 ± 0.07 ^b^	0.286 ± 0.001 ^c^	——	——
J3-starch	3.15 ± 0.02 ^d^	10.22 ± 0.02 ^d^	16.57 ± 0.02 ^c^	0.318 ± 0.002 ^d^	——	——
J4-starch	2.37 ± 0.04 ^e^	8.85 ± 0.06 ^e^	14.93 ± 0.26 ^d^	0.368 ± 0.006 ^e^	——	——
J5-starch	2.08 ± 0.0.06 ^f^	7.70 ± 0.04 ^f^	13.34 ± 0.05 ^e^	0.403 ± 0.005 ^f^	——	——
OSA-starch	5.89 ± 0.09 ^a^	13.48 ± 0.02 ^a^	20.09 ± 0.01 ^a^	0.238 ± 0.002 ^a^	0.0170 ± 0.0007 ^a^	73.50 ± 0.90 ^d^
J1-OSA-starch	4.78 ± 0.01 ^b^	12.57 ± 0.05 ^b^	19.26 ± 0.07 ^b^	0.265 ± 0.001 ^b^	0.0172 ± 0.0004 ^b^	74.30 ± 0.90 ^d^
J2-OSA-starch	4.14 ± 0.09 ^c^	11.70 ± 0.15 ^c^	18.35 ± 0.08 ^b^	0.284 ± 0.003 ^c^	0.0175 ± 0.0005 ^c^	75.60 ± 0.40 ^c^
J3-OSA-starch	3.15 ± 0.01 ^d^	10.23 ± 0.01 ^d^	16.59 ± 0.04 ^c^	0.318 ± 0.001 ^d^	0.0179 ± 0.0002 ^d^	77.40 ± 0.90 ^b^
J4-OSA-starch	2.41 ± 0.06 ^e^	8.94 ± 0.13 ^e^	15.19 ± 0.37 ^d^	0.363 ± 0.007 ^e^	0.0183 ± 0.0006 ^e^	79.10 ± 0.90 ^b^
J5-OSA-starch	2.12 ± 0.05 ^f^	7.75 ± 0.06 ^f^	13.37 ± 0.04 ^e^	0.399 ± 0.006 ^f^	0.0186 ± 0.0003 ^f^	80.40 ± 1.30 ^a^

**Note:** Values in the table are mean ± standard deviation. Different subscript letters within the same column indicate significant differences (*p* < 0.05).

**Table 2 foods-11-02893-t002:** Surface elemental composition of J-starch and J-OSA-starch.

Samples	Native Starch	J1	J2	J3	J4	J5	OSA-Starch	J1-OSAh	J2-OSA	J3-OSA	J4-OSA	J5-OSA
C (%)	56.50	57.03	57.18	56.77	56.32	56.87	56.9	57.00	57.60	57.90	58.40	58.60
O (%)	42.70	42.26	42.10	42.53	42.95	42.41	42.1	41.80	41.20	41.00	40.50	40.20
Na (%)	0	0	0	0	0	0	0.23	0.30	0.37	0.43	0.49	0.56
N/(%)	0.72	0.71	0.72	0.70	0.73	0.72	0.95	1.08	1.09	1.02	0.99	1.12

**Table 3 foods-11-02893-t003:** DSC analysis of the native and treated corn starch samples.

Samples	T_0_ (°C)	T_P_ (°C)	T_C_ (°C)	∆H (J/g)
Native starch	62.58 ± 0.03 ^a^	68.42 ± 0.04 ^a^	89.12 ± 0.23 ^a^	17.02 ± 0.09 ^a^
J1-starch	56.48 ± 0.05 ^c^	68.44 ± 0.06 ^a^	83.15 ± 0.27 ^b^	9.80 ± 0.12 ^b^
J2-starch	61.44 ± 0.11 ^b^	66.77 ± 0.07 ^b^	83.61 ± 0.16 ^c^	8.81 ± 0.13 ^c^
J3-starch	55.83 ± 0.18 ^d^	65.19 ± 0.02 ^c^	81.80 ± 0.21 ^d^	5.65 ± 0.09 ^d^
J4-starch	53.81 ± 0.20 ^e^	61.45 ± 0.04 ^d^	77.65 ± 0.16 ^e^	2.52 ± 0.14 ^e^
J5-starch	52.74 ± 0.05 ^f^	61.4 ± 0.06 ^d^	74.63 ± 0.21 ^f^	1.89 ± 0.13 ^f^
OSA-starch	61.88 ± 0.03 ^a^	67.51 ± 0.03 ^a^	88.34 ± 0.05 ^a^	16.78 ± 0.03 ^a^
J1-OSA-starch	61.53 ± 0.03 ^b^	67.35 ± 0.04 ^a^	86.95 ± 0.33 ^b^	16.15 ± 0.07 ^b^
J2-OSA-starch	61.44 ± 0.09 ^b^	67.10 ± 0.01 ^b^	83.34 ± 0.12 ^c^	14.37 ± 0.28 ^c^
J3-OSA-starch	60.75 ± 0.18 ^c^	67.01 ± 0.02 ^c^	82.86 ± 0.21 ^d^	12.98 ± 0.18 ^d^
J4-OSA-starch	59.77 ± 0.18 ^d^	66.81 ± 0.06 ^d^	80.92 ± 0.15 ^e^	11.97 ± 0.11 ^e^
J5-OSA-starch	59.40 ± 0.05 ^d^	65.98 ± 0.08 ^e^	77.63 ± 0.11 ^f^	10.60 ± 0.22 ^f^

**Note:** Values in the table are mean ± standard deviation. Different subscript letters within the same column indicate significant differences (*p* < 0.05).

**Table 4 foods-11-02893-t004:** RVA, EA, and ESI analysis of the native starch and J-OSA-starch samples.

	Parameters	Native Starch	OSA-Starch	J1-OSA-Starch	J2-OSA-Starch	J3-OSA-Starch	J4-OSA-Starch	J5-OSA-Starch
RVA	PV (cp)	5149 ± 12 ^e^	8991 ± 15 ^a^	7771 ± 13 ^b^	6223 ± 9 ^c^	5156 ± 15 ^d^	4541 ± 16 ^f^	4012 ± 10 ^g^
	TV (cp)	4013 ± 16 ^d^	5101 ± 12 ^a^	4577 ± 9 ^b^	4263 ± 7 ^c^	3626 ± 13 ^e^	3043 ± 5 ^f^	2413 ± 11 ^g^
	FV (cp)	4817 ± 5 ^d^	6077 ± 7 ^a^	5503 ± 14 ^b^	5072 ± 12 ^c^	4389 ± 18 ^e^	3745 ± 11 ^f^	3021 ± 9 ^g^
	BD (cp)	1136 ± 12 ^g^	3890 ± 14 ^a^	3194 ± 19 ^b^	1960 ± 9 ^c^	1530 ± 15 ^d^	1498 ± 12 ^e^	1599 ± 13 ^f^
	SB(cp)	804 ± 14 ^c^	976 ± 10 ^a^	926 ± 8 ^b^	809 ± 6 ^c^	763 ± 12 ^d^	702 ± 11 ^e^	608 ± 14 ^f^
	PT (°C)	74.25 ± 0.50 ^a^	73.14 ± 0.70 ^b^	72.20 ± 0.30 ^c^	71.04 ± 0.20 ^d^	70.10 ± 0.80 ^e^	68.55 ± 0.70 ^f^	66.65 ± 0.80 ^g^
EA and ESI	EA	0.26 ± 0.02 ^a^	0.46 ± 0.01 ^b^	0.54 ± 0.01 ^c^	0.62 ± 0.02 ^d^	0.69 ± 0.01 ^e^	0.73 ± 0.01 ^f^	0.82 ± 0.01 ^g^
	ESI (min)	92.5 ± 10.2 ^g^	152.8 ± 9.1 ^f^	209.2 ± 12.5 ^e^	264.9 ± 15.8 ^d^	303.3 ± 20.5 ^c^	352.7 ± 22.4 ^b^	410.1 ± 21.6 ^a^

**Note:** Values in the table are mean ± standard deviation. Different subscript letters within the same column indicate significant differences (*p* < 0.05).

## Data Availability

The data presented in this study are available on request from the corresponding author.

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
