# Peer review of "Effects of Jet Milling Pretreatment and Esterification with Octenyl Succinic Anhydride on Physicochemical Properties of Corn Starch"

_foods, 2022, doi:10.3390/foods11182893_

Round 1
Reviewer 1 Report
The manuscript presented properties of corn starch before and after jet milling and esterification. The strong point is intensive analysis data of corn starch properties whereas the weak point is not much discussion to support the finding. Some points have to be clarified.
1. Abstract, authors used ‘depending of the classifier’s rotational speed’ many times. Please revise the sentences to keep abstract compact and informative.
2. Page 3 Line 88 “…….short pulverization…” and “…….low pulverization temperature….”
Please add more detail of time and temperature of pulverization.
3. Please state pulverization temperature used in this study in the section “Jet milling pretreatment”.
4. How much is temperature (of samples) increase after jet milling?
5. Page 6, transparency measurement explained in section 2.11 was done at 100C, but the results in Figure 6 showed at 3 temperatures. Please revise.
6. Section 3 Results should be changed to Results and Discussion. Then please add more intensive discussion to support the finding. Authors can consider using the finding in sections 3.1-3.5 to deeply explain and discuss the properties in sections 3.6-3.9.
7. Table 1 should present the values of normal corn starch to determine the effect of OSA and jet milling speed.
8. Page 9 Line 305 “…..mechanical force of jet milling destroyed the internal crystalline structure of corn starch…..”
Please explain more detail how mechanical force change the crystalline structure.
9. For transparency, when starch is fully gelatinized at 100C, why did light transmittance of all samples still show significant difference (Figure 6). Please explain more detail.
10. Content presenting in section 4 Discussion should be ‘Conclusion’.
Author Response
Author’s responses to reviewer’s comments
Reviewer
Comments to the Author
The manuscript presented properties of corn starch before and after jet milling and esterification. The strong point is intensive analysis data of corn starch properties whereas the weak point is not much discussion to support the finding. Some points have to be clarified.
We gratefully appreciate your time, patience, and helpful comments regarding the manuscript. Please find below our responses to your comments, one by one.
Minor comments -
-1. Abstract, authors used ‘depending of the classifier’s rotational speed’ many times. Please revise the sentences to keep abstract compact and informative.
-Thanks for your comments, this section has been revised. Please find the changes texted in red colour.
-2. Page 3 Line 88 “…….short pulverization…” and “…….low pulverization temperature….”
-Thank you for your comments, we have supplement the crushing time and temperature conditions in the description.
-3. Please state pulverization temperature used in this study in the section “Jet milling pretreatment”
-According to your suggestion, we have supplement the grinding time and temperature conditions in 2.2 Materials and methods.
-4. How much is temperature (of samples) increase after jet milling?
-In the process of air pulverization of starch, the temperature rise is very low, and the flow of air flow will quickly take away the heat generated by mechanical collision. We measure the temperature rise at about 4 °C.
-5. Page 6, transparency measurement explained in section 2.11 was done at 100C, but the results in Figure 6 showed at 3 temperatures. Please revise.
-You are right. We have supplement grinding time and temperature conditions in 2.11 Materials and methods.
-6. Section 3 Results should be changed to Results and Discussion. Then please add more intensive discussion to support the finding. Authors can consider using the finding in sections 3.1-3.5 to deeply explain and discuss the properties in sections 3.6-3.9
-Thanks for your comments. Based on your suggestion in this section, we have made changes. Please find the changes texted in red colour.
-7. Table 1 should present the values of normal corn starch to determine the effect of OSA and jet milling speed.
-According to your suggestion, the data for native and pretreated starches are supplemented in Table 1 based on your comments.
-8. Page 9 Line 305 “…..mechanical force of jet milling destroyed the internal crystalline structure of corn starch…..”
-According to your suggestion, we have explained it in more detail in this part of the text.
-9. For transparency, when starch is fully gelatinized at 100C, why did light transmittance of all samples still show significant difference (Figure 6). Please explain more detail.
-Considering your advice, these phenomena may be attributed to the destruction of the crystalline structure and transformation to amorphous form. Because of that, the water can enter to the interior of starch granules, leading to increased transparency which reduced the molecular weight of starch and increased the number of small molecules, reducing light scattering and refraction, thereby changing its transparency.
-10. Content presenting in section 4 Discussion should be ‘Conclusion’.
-According to your suggestion. Make changes to this section based on your comments in Results and Discussion.
Once again, thank you for your time, patience, and helpful advices and we hope our revision make you satisfy.
Sincerely
The authors
Reviewer 2 Report
Comments to authors
This manuscript “Effects of Jet Milling Pre-treatment and Esterification with Oc-2 tenyl Succinic Anhydride on Structural and Physicochemical 3 Properties of Corn Starch” is exciting and has potential utility in deciding the role of modified starch in the preparation of various food products. Some questions in the 'comments to the authors' section need clarification. These are just a bit more than 'minor revisions', so I must select 'major revisions' for this first recommendation.
1. Abstract: What is the purpose? What is the conclusion?. Rewrite this section.
2. Also, briefly describe the Jet milling in introduction section.
3. At the end of the introduction, justify why pre-treatment is required while degree of substitution is affected by OSA concentration, time, and temperature.
4. In the introduction section, please mention the limits of OSA for modification of starch for food grade. According to CFR, OSA is permitted 3% for many countries. Why you select this concentration.
5. Figure 1. Add the full name of the sample with figure caption and add figure Jet milled starch with native so that it can be concluded that changes are due to OSA or Jet milling.
6. Re-plot figure 5; start values from 40; as the gelatinization process starts after 40, you should focus on DSC peaks.
7. In pasting properties, first PV increase then decrease; explain this and add the possible reason.
8. It is also suggested that in all results, add results for pre-treated starch with native starch.
9. Conclusion is not informative; rewrite this section.
Author Response
Author’s responses to reviewer’s comments
Reviewer
Comments to the Author
This manuscript “Effects of Jet Milling Pre-treatment and Esterification with Oc-2 tenyl Succinic Anhydride on Structural and Physicochemical 3 Properties of Corn Starch” is exciting and has potential utility in deciding the role of modified starch in the preparation of various food products. Some questions in the 'comments to the authors' section need clarification. These are just a bit more than 'minor revisions', so I must select 'major revisions' for this first recommendation.
We gratefully appreciate your time, patience, and helpful comments regarding the manuscript. Please find below our responses to your comments, one by one.
Minor comments -
-1. Abstract: What is the purpose? What is the conclusion?. Rewrite this section.
- According to your valuable advice, the abstract has been rewritten. Please read it once again and find the modifications texted.
- 2. Also, briefly describe the Jet milling in introduction section.
-Considering your advice, we have supplemented the principle of jet milling in the introduction section. Please find the modifications texted in red colour.
-3. At the end of the introduction, justify why pre-treatment is required while degree of substitution is affected by OSA concentration, time, and temperature.
-You are right. According to your valuable suggestion, the description of this section has been added. Please find it texted in red colour in the introduction.
- 4. In the introduction section, please mention the limits of OSA for modification of starch for food grade. According to CFR, OSA is permitted 3% for many countries. Why you select this concentration.
- Considering your advice, the limits of OSA for modification of starch for food grade had been added in the introduction section. Explain to you here, the United Nations Food and Agriculture Organization and the World Health Organization (FAO/WHO) stipulated that the OSA ester could be used in food, but the maximum processing capacity of octenyl succinic anhydride should not be exceeding 3% level, and the influencing factors such as temperature and pH are sufficient in the esterification modification of different starches.
- 5. Figure 1. Add the full name of the sample with figure caption and add figure Jet milled starch with native so that it can be concluded that changes are due to OSA or Jet milling.
- Based on your comments, we have added sample full name and figure description in Figure 1, and added figure Jetmilled starch with native. Please find the modifications texted in Figure.
-6. Re-plot figure 5; start values from 40; as the gelatinization process starts after 40, you should focus on DSC peaks.
-According to your suggestion, we adjusted the X-axis range to 40℃~90℃, it can be seen that the quality of the pictures has been greatly improved. Please find the modifications texted in Figure 5.
-7. In pasting properties, first PV increase then decrease; explain this and add the possible reason
-Considering your advice, we have added the possible reason why the first PV increase then decrease in pasting properties section. Here we show you this reason, the PV value of OSA starch increases relative to native starch, due to the introduction of octenyl succinic anhydride, the starch molecules with the same negative charge would repel each other.However, the crystallinity of the starch pretreated by air flow gradually decreases with the increase of the classification speed, which leads to a trend of decreasing trend of the samples after OSA esterification.
-8. It is also suggested that in all results, add results for pre-treated starch with native starch.
-According to your suggestion. Adding the indicators of jet milled starch can increase the comparison effect after esterification. We have added these results to the corresponding position of the article, please check.
-9. Conclusion is not informative; rewrite this section.
-Thank you for your valuable comments. Based on your comments, we have reorganized the conclusion part.
Once again, thank you for your time, patience, and helpful advices and we hope our revision make you satisfy.
Sincerely
The authors
Reviewer 3 Report
Structural and Physicochemical Properties of Corn Starch
R:// Structural is part of physicochemical properties.
Starch is a natural polymer composed of anhydrous glucose units linked by glyco- 68 sidic bonds
R:// Starch in my opinión is not a polymer, it is a micro or submicron particle formed by amylose, amylopectin, fat, proteins, minerals, salts, and nano crystals (orthorhombic and/or hexagonal). Then if your goal is about modification, the central point is to determine the interaction with these elements because of modifications.
X-ray analysis
R: Please separated X-ray and IR images, and 2ÆŸ (Degrees)
I believe that your x-ray patterns can be filtered for a better analysis
Fig. 2-A shows XRD spectra for the native and treated corn starch.
R:// These are in fact, X-ray patterns, because the wavelength of Cu is 1.5406 Anstrons, and you are not using a polychromatic source, please change it.
The native corn 300 starch showed a typical "A" type structure
R:/ A-type is not a structure, A-type starch contains orthorhombic crystalline structure; B-type starch has an hexagonal structure, and C-type starch has both orthorhombic and hexagonal structure.
with strong characteristic diffraction peaks 301 around 15.8°, 17.1°, 18.2°, and 23.5°
R:// Each one of the crystalline structures present in starch were indexed by Rodriguez-Garcia et al. 2021. The intensity is directly related to the form and structure factors, and also depends on the external parameters.
The degree of crystallinity for the J-OSA starch samples de- 303 creased as the jet milling classifier speed increased
R:// By a detailed analysis of your patterns, it is not possible to see a decrease in the crystalline quality.
These results indicate that jet milling pretreatment significantly reduced the 308 crystallinity of corn starch and enhanced the esterification of corn starch with OSA.
R:// It is necessary to do the calculation of the crystalline percent.
. Jet milling pretreatment significantly changed gelatinization properties
R:// What kind of gelatinization properties?
What is the origin of teh gelatinization?
h. From Table 3, values of the paste characteristics of the native corn starch are high, 378
R:// You're describing thermal properties, this sentence is for the next section.
The endothermic transition shown in these figures corresponds to the gelatinization, in which an order-disorder transition takes place. According toGong et al 2016, and King et al 2012, the crystalline structures present in starch are orthorhombic and hexagonal nanocrystals. Esquivel-Fajardo et al 2022, pointed out that this transition corresponds to the crystal solvatation. THe analysis of your thermal properties must be improved.
According to the X-ray patterns, J5-Starch exhibits a crystalline structure, but DSC does not exhibit the same behaviours, is it possible to have a mistake during the DSC experiments? Please check it.
the energy required to break the double helix structure of the starch granules, resulting in 384 easier pasting of the starch, as well as low pasting temperature and enthalpy value.
R:// Again, this sentence is part of the next section. On the other hand, could you indicate if you have the information about the theory or experiments in which the hexagonal or orthorhombic structures form a double hélix (Naegeli Amylodextrin and Its Relationship to Starch Granule Structure. 11. Role of Water in Crystallization of B-Starch* BIOPOLYMERS VOL. 11, 2241-2250 (19’72). However, they never proved the existence of a double hélix.
R:/7 It is necessary to include a Figure for pasting profile.
This can be attributed to the 391 introduction of macromolecular OSA groups into the starch after starch modification by 392 OSA, which increased the relative molecular weight of the starch,
R:// Starch is formed by amylose, amylopectin, fat, proteins, wáter, ions, and/or hexagonal and orthorhombic nanocrystals, what is the meaning of molecular weight of starch, or are you referring to the molecular weight of amylose and amylopectin?
*Please improve the quality of each one of your images
Fig. 5 from 30 to 90 °C can improve the image.
Gong, B., Liu, W., Tan, H., Yu, D., Song, Z., & Lucia, L. A. (2016). Understanding shape and morphology of unusual tubular starch nanocrystals. Carbohydrate Polymers, 151, 666–675. https://doi.org/10.1016/j.carbpol.2016.06.010
Rodriguez-Garcia, M. E., Hernandez-Landaverde, M. A., Delgado, J. M., Ramirez-Gutierrez, C. F., Ramirez-Cardona, M., Millan-Malo, B. M., & Londoño-Restrepo, S. M. (2020). Crystalline Structures of the main components of Starch. Current Opinion in Food Science. https://doi.org/10.1016/j.cofs.2020.10.002
Discusion, must be base don each one of the experiments and their correlations, plese define waht kind of physicochemical changes are produced in starch due to your modifications. The paper is interesting but looks like a technical report.
Author Response
Author’s responses to reviewer’s comments
Reviewer
We gratefully appreciate your time, patience, and helpful comments regarding the manuscript. Please find below our responses to your comments, one by one.
Minor comments -
-1. Structural and Physicochemical Properties of Corn Starch
R:// Structural is part of physicochemical properties.
-Thank you for your valuable comments, we have revised the title to "Effects of Jet Milling Pretreatment and Esterification with Octenyl Succinic Anhydride on Physicochemical Properties of Corn Starch".
-2. Starch is a natural polymer composed of anhydrous glucose units linked by glyco- 68 sidic bonds
R:// Starch in my opinión is not a polymer, it is a micro or submicron particle formed by amylose, amylopectin, fat, proteins, minerals, salts, and nano crystals (orthorhombic and/or hexagonal). Then if your goal is about modification, the central point is to determine the interaction with these elements because of modifications.
-Thank you for the detailed instructions for this section, we have corrected the concepts in the text. Please find it texted in red colour in the manuscript.
-3. X-ray analysis
R: Please separated X-ray and IR images, and 2ÆŸ (Degrees)
-Thanks, based on your comments, we have placed the FTIR image in the appropriate place in the text.
-4. I believe that your x-ray patterns can be filtered for a better analysisFig. 2-A shows XRD spectra for the native and treated corn starch.
R:// These are in fact, X-ray patterns, because the wavelength of Cu is 1.5406 Anstrons, and you are not using a polychromatic source, please change it.
-You are right. Considering your advice, we have revised this part of the description in 2.6.1. Please read it once again and find the modifications texted in red colour.
-5 The native corn 300 starch showed a typical "A" type structure
R:/ A-type is not a structure, A-type starch contains orthorhombic crystalline structure; B-type starch has an hexagonal structure, and C-type starch has both orthorhombic and hexagonal structure.
-Thanks for your explanation of this concept, we have corrected the relevant part of the text to "A-type starch".
-6. With strong characteristic diffraction peaks 301 around 15.8°, 17.1°, 18.2°, and 23.5°
R:// Each one of the crystalline structures present in starch were indexed by Rodriguez-Garcia et al. 2021. The intensity is directly related to the form and structure factors, and also depends on the external parameters.
-Thank you for your valuable comments. In our measurement process, all samples use the same measurement conditions to analyze the difference between J-starch and J-OSA-starch.
-7. The degree of crystallinity for the J-OSA starch samples de- 303 creased as the jet milling classifier speed increased
R:// By a detailed analysis of your patterns, it is not possible to see a decrease in the crystalline quality.
-Considering your advice, we have supplemented the XRD pattern after jet mill pretreatment and annotated relative crystallinity.
-8. These results indicate that jet milling pretreatment significantly reduced the 308 crystallinity of corn starch and enhanced the esterification of corn starch with OSA.
R:// It is necessary to do the calculation of the crystalline percent.
-Thanks for your opinion, we have adopted jade6 software to calculate the crystallinity and marked it in the picture.
-9. Jet milling pretreatment significantly changed gelatinization properties
R:// What kind of gelatinization properties?
-The gelatinization characteristics mainly include changes in the peak viscosity, valley viscosity, final viscosity, retrogradation value, decay value, and gelatinization starting temperature of starch. Considering your advice,we have made changes in the text.
-10. What is the origin of teh gelatinization?
-Upon inspection, the RVA-Super4 was used to determine the gelatinization characteristic value. A 3.5 g starch sample was dispersed in 30 mL of distilled water, and the instrument automatically heated at a uniform speed to gelatinize the starch.
-11. From Table 3, values of the paste characteristics of the native corn starch are high, 378
R:// You're describing thermal properties, this sentence is for the next section.
-Thanks for your opinion.This description has been changed based on your suggestion.Please find the changes texted in red colour.
-12. The endothermic transition shown in these figures corresponds to the gelatinization, in which an order-disorder transition takes place. According toGong et al 2016, and King et al 2012, the crystalline structures present in starch are orthorhombic and hexagonal nanocrystals. Esquivel-Fajardo et al 2022, pointed out that this transition corresponds to the crystal solvatation. THe analysis of your thermal properties must be improved.
According to the X-ray patterns, J5-Starch exhibits a crystalline structure, but DSC does not exhibit the same behaviours, is it possible to have a mistake during the DSC experiments? Please check it.
-Thank you for your reminder, we re-checked the data, re-plotted the DSC plot, and zoomed in. It can be seen that the J5-OSA-starch shows a small heat absorption peak.
-13. the energy required to break the double helix structure of the starch granules, resulting in 384 easier pasting of the starch, as well as low pasting temperature and enthalpy value.
R:// Again, this sentence is part of the next section. On the other hand, could you indicate if you have the information about the theory or experiments in which the hexagonal or orthorhombic structures form a double hélix (Naegeli Amylodextrin and Its Relationship to Starch Granule Structure. 11. Role of Water in Crystallization of B-Starch* BIOPOLYMERS VOL. 11, 2241-2250 (19’72). However, they never proved the existence of a double hélix.
-You are right. According to your valuable advice, this description has been reorganized based on your comments. Please find the changes texted in red colour.
-14. R:/7 It is necessary to include a Figure for pasting profile.
-I am very sorry, because the school is currently in the epidemic control stage, the experimental raw materials are closed in the laboratory, and supplementing this part requires re-experimentation to supplement the curve data.
-15. This can be attributed to the 391 introduction of macromolecular OSA groups into the starch after starch modification by 392 OSA, which increased the relative molecular weight of the starch,
R:// Starch is formed by amylose, amylopectin, fat, proteins, wáter, ions, and/or hexagonal and orthorhombic nanocrystals, what is the meaning of molecular weight of starch, or are you referring to the molecular weight of amylose and amylopectin?
-Yes, we are referring to the molecular weight of amylose and amylopectin.
-16. *Please improve the quality of each one of your images
-Thanks for your comments, all images have been redrawn based on your comments.
-17. Fig. 5 from 30 to 90 °C can improve the image.
-Considering your advice, the abscissa of the image has been redrawn based on your comments.
Once again, thank you for your time, patience, and helpful advices and we hope our revision make you satisfy.
Sincerely
The authors

Round 2
Reviewer 1 Report
No more comment.
Reviewer 2 Report
Authors address all comments suggested by me.